# The Microbiome as a Potential Target for Therapeutic Manipulation in Pancreatic Cancer

**DOI:** 10.3390/cancers13153779

**Published:** 2021-07-27

**Authors:** Rozana Abdul Rahman, Angela Lamarca, Richard A. Hubner, Juan W. Valle, Mairéad G. McNamara

**Affiliations:** 1Experimental Cancer Medicine Team, The Christie NHS Foundation Trust, Manchester M20 4BX, UK; rozana.rahman@nhs.net; 2Department of Medical Oncology, The Christie NHS Foundation Trust/Division of Cancer Sciences, University of Manchester, Manchester M20 4BX, UK; angela.lamarca@nhs.net (A.L.); richard.hubner@nhs.net (R.A.H.); 3Division of Cancer Sciences, University of Manchester/Department of Medical Oncology, The Christie NHS Foundation Trust, Manchester M20 4BX, UK; juan.valle@nhs.net

**Keywords:** microbiome, pancreatic ductal adenocarcinoma (PDAC), therapeutic manipulation

## Abstract

**Simple Summary:**

Pancreatic cancer is one of the most lethal cancers. It is a difficult cancer to treat, and the complexity surrounding the pancreatic tumour is one of the contributing factors. The microbiome is the collection of microorganisms within an environment and its genetic material. They reside on body surfaces and most abundantly within the human gut in symbiotic balance with their human host. Disturbance in the balance can lead to many diseases, including cancers. Significant advances have been made in cancer treatment since the introduction of immunotherapy, and the microbiome may play a part in the outcome and survival of patients with cancer, especially those treated with immunotherapy. Immunotherapy use in pancreatic cancer remains challenging. This review will focus on the potential interaction of the microbiome with pancreas cancer and how this could be manipulated.

**Abstract:**

Pancreatic ductal adenocarcinoma (PDAC) is one of the most lethal cancers and is projected to be the second most common cause of cancer-related death by 2030, with an overall 5-year survival rate between 7% and 9%. Despite recent advances in surgical, chemotherapy, and radiotherapy techniques, the outcome for patients with PDAC remains poor. Poor prognosis is multifactorial, including the likelihood of sub-clinical metastatic disease at presentation, late-stage at presentation, absence of early and reliable diagnostic biomarkers, and complex biology surrounding the extensive desmoplastic PDAC tumour micro-environment. Microbiota refers to all the microorganisms found in an environment, whereas microbiome is the collection of microbiota and their genome within an environment. These organisms reside on body surfaces and within mucosal layers, but are most abundantly found within the gut. The commensal microbiome resides in symbiosis in healthy individuals and contributes to nutritive, metabolic and immune-modulation to maintain normal health. Dysbiosis is the perturbation of the microbiome that can lead to a diseased state, including inflammatory bowel conditions and aetiology of cancer, such as colorectal and PDAC. Microbes have been linked to approximately 10% to 20% of human cancers, and they can induce carcinogenesis by affecting a number of the cancer hallmarks, such as promoting inflammation, avoiding immune destruction, and microbial metabolites can deregulate host genome stability preceding cancer development. Significant advances have been made in cancer treatment since the advent of immunotherapy. The microbiome signature has been linked to response to immunotherapy and survival in many solid tumours. However, progress with immunotherapy in PDAC has been challenging. Therefore, this review will focus on the available published evidence of the microbiome association with PDAC and explore its potential as a target for therapeutic manipulation.

## 1. Introduction

Pancreatic ductal adenocarcinoma (PDAC) is one of the most lethal cancers. It is the 4th most frequent cause of cancer-related death worldwide and is projected to be the second most common cause of cancer-related death by 2030 [1]. While the overall cancer survival rates have shown moderate improvement with multimodality treatment, the overall 5-year survival rate for patients diagnosed with PDAC remains relatively stagnant, between 7% and 9% for the past three decades.

Despite recent improvements in surgical, chemotherapy, and radiotherapy techniques, pancreatic cancer, even when initially resectable, remains a poor prognosis disease. This is attributable to subclinical metastatic disease at presentation, early recurrence, late-stage at presentation, lack of effective treatments available, and absence of a biomarker that can detect this cancer in its early or pre-invasive form. Moreover, pancreatic inflammation is considered a long-term risk factor, and chronic pancreatitis can increase the risk of PDAC up to 20 times [2]. Poor prognosis is also attributed to the complex biology surrounding the extensive desmoplastic PDAC tumour micro-environment, leading to hypovascularity, hypoxia, poor drug delivery, and ineffective therapies [3,4]. Moreover, despite significant advances with immunotherapy for the treatment of many cancers, the progress of immunotherapy in PDAC has been challenging.

Microbes have been linked to approximately 10–20% of human cancers and are able to induce carcinogenesis [5,6,7]. The microbiome signature is potentially linked to response to therapy, including immunotherapy, and survival in many solid tumours. Therefore, this review will focus on the available evidence on the microbiome in PDAC and address its potential as a target for therapeutic manipulation.

## 2. Therapeutic Challenges in Treating Patients with Pancreatic Cancer

Pancreatic ductal adenocarcinoma is one of the most stroma-rich cancers, accounting for 50–80% of tumour volume. The PDAC stroma supports tumour growth, promotes metastases, and simultaneously serves as a physical barrier to drug delivery and is highly resistant to conventional therapies [3,4,8]. Another feature of the pancreatic cancer tumour environment is immune evasion or restriction of immune surveillance, whereby the PDAC microenvironment is composed of T regulatory cells (Treg), tumour-associated macrophages (TAMs), and myeloid-derived suppressive cells (MDSCs), which block CD8^+^ T-cell roles in tumour recognition and clearance [9].

Difficulty in early detection of PDAC with the lack of biomarkers and accurate diagnostic radiology tests further contributes to the challenges in treating patients with pancreatic cancer [10,11,12]. DAC, even when thought to be diagnosed at early stages, may already have subclinical metastases on currently available imaging modality [13,14]. Recent developments in theranostic nuclear imaging will hopefully improve diagnostic ability [12].

### 2.1. Surgery and Chemotherapy in Patients with Pancreas Cancer

Surgical resection offers the only hope for a cure in 20–30% of patients with loco-regional disease [15]. The best outcome is achieved when surgery is followed by adjuvant modified FOLFIRINOX (mFOLFIRINOX) chemotherapy; with a median OS of 54.4 months vs. 35 months with gemcitabine in the randomised study by Conroy et al. [16]. This contrasts with OS of 11.1 months for patients with metastatic disease receiving FOLFIRINOX chemotherapy [17]. FOLFIRINOX is mainly used among the fittest patients both in the adjuvant and advanced settings.

European Study Group for Pancreatic Cancer-4 (ESPAC-4) was a phase III study that showed the best OS outcome in the adjuvant setting prior to mFOLFIRINOX data. The study randomised 722 patients to receive either adjuvant gemcitabine alone or gemcitabine with oral capecitabine (GEM/CAP). Median survival for patients treated with GEM/CAP was 28.0 months (95% confidence interval (CI), 23.5–31.5) and 25.5 months (95% CI, 22.7–27.9) for gemcitabine alone (hazard ratio (HR) 0.82 (95% CI, 0.68–0.98); and *p* = 0.032).

MPACT was another pivotal phase III study for the treatment of PDAC in the metastatic setting using gemcitabine and nab-paclitaxel. Median OS was 8.5 months in the nab-paclitaxel-gemcitabine group as compared with 6.7 months in the gemcitabine group (HR for death, 0.72; 95% CI, 0.62 to 0.83; *p* < 0.001) [18].

In patients who are less fit, single-agent gemcitabine that has been in use for the past three decades is still utilised, both in the adjuvant and metastatic settings. Novel therapeutics that can improve survival for these patients are desperately needed.

### 2.2. Immunotherapy Use in Pancreatic Cancer and the Associated Challenges

The advent of immune checkpoint inhibitors (ICI) such as anti-cytotoxic T-lymphocyte-associated protein 4 (CTLA4), anti-programmed death protein-1(PD-1), and anti-PD-1 ligand 1 (PD-L1) has revolutionised cancer treatment in the last decade. However, the progress in using immunotherapy in patients with PDAC remains challenging.

Following success in melanoma, ipilimumab (anti-CTLA4 antibody) was the first ICI to be evaluated in PDAC. A phase II trial in 2010 utilised ipilimumab for 27 patients with PDAC, seven with localised disease, and 20 with metastatic disease. There was no objective response [19]. A phase II study using anti-PD-L1 antibody durvalumab in 32 previously treated patients with metastatic PDAC also showed no objective response [20]. However, a phase II study of the combinations of PD-L1 and CTLA-4 inhibitors (durvalumab and tremelimumab) for four cycles, followed by durvalumab maintenance for a year, resulted in an objective response rate (ORR) of 3.5% and greater grade 3 or higher adverse events, but did not progress into further phases [20]. Emerging data have demonstrated that cytotoxic chemotherapy may potentiate the effects of ICIs in many cancer types [21]. A phase II study examined the combination of gemcitabine, nab-paclitaxel with durvalumab and tremelimumab as first-line treatment for patients with metastatic PDAC and initially demonstrated a disease control rate of 100%, and 73% of patients had a partial response [22]. However, a subsequent randomised phase II study failed to show a difference in progression-free survival (PFS), OS, or response rate between the groups; gemcitabine and nab-paclitaxel, with or without durvalumab and tremelimumab [23].

Nevertheless, mismatch repair deficiency was approved by the U.S. Food and Drug Administration (FDA) as tissue agnostic for pembrolizumab therapy for microsatellite instability-high or deficient mismatch repair (MSI-H/dMMR) solid tumours in 2017. This is based on the phase II KEYNOTE-158 study involving 22 patients with advanced PDAC, among 233 patients with advanced and previously treated non-colorectal cancer, with a reported ORR of 18.2% and a median PFS and OS of 2.1 months and 4 months, respectively [24]. However, the MSI-H/dMMR PDAC phenotype is relatively uncommon and ranges between 0.8% and 2% of PDACs [25]. Moreover, the ORR of 18.2% seen in MSI-H PDAC in the KEYNOTE 158 trial was substantially lower than the response rates seen in MSI-H cholangiocarcinoma (40.9%), small intestine (42.1%), gastric (45.8%) and endometrial cancers (57.1%) [24].

One of the earlier immunotherapies investigated in PDAC was the GVAX vaccine, a heterologous whole-cell vaccine composed of two irradiated allogeneic PDAC cell lines modified to secrete GM-CSF to induce the recruitment of immune cells, including effector T cells [26]. GVAX showed positive results in a phase I and phase II study in combination with cyclophosphamide [26,27]. The GVAX vaccine in combination with ipilimumab was studied in a phase 1B trial, either alone or in combination with GVAX in 30 previously treated patients with advanced PDAC [28]. The GVAX/ipilimumab arm had a median OS of 5.7 months compared to 3.6 months with ipilimumab monotherapy, with a similar disease control rate in both arms. Several trials are currently ongoing in the treatment of PDAC, combining GVAX and anti-PD-1 antibody treatment (NCT02451982; NCT02648282).

The relative lack of success of immunotherapy in the treatment of patients with PDAC is not fully understood but may in some way be attributable to the patient microbiome.

## 3. What Is Microbiome?

Microbiome and microbiota are two terms with subtle differences that are often used interchangeably. Microbiota refers to all the microorganisms found in an environment, including bacteria, archaea, viruses and fungi. In contrast, the microbiome refers to the collection of microorganisms and their genome within an environment [29]. Metagenomics is the study of genetic material found within an environmental sample. Body surfaces and mucosal layers, including skin, nasal, oral and vaginal cavities, are all inhabited by various microbes, but they are the most diverse and abundantly found within the distal gut. The human gut microbiota typically includes up to 10^13^–10^14^ organisms, similar to the number of human cells, and the human microbiome consists of a unique genome of up to 3 × 10^6^ genes [30].

Most organisms with a gut have a microbiome that co-evolved in symbiosis with the host [31]. It develops throughout life after birth [32,33]. The microbiome is involved in basic human biological processes, including modulating metabolic activities, regulating epithelial development, and influencing innate immunity [34]. In terms of nutrition, the microbiota can increase energy extraction from food, increase nutrient harvest [35,36,37] and alter appetite signalling [35,36,38,39]. Furthermore, the microbiota is also involved in the metabolism of undigested carbohydrates, the synthesis of amino acids, and the biosynthesis of vitamins [35,40]. The human microbiota is also vital in providing a physical barrier to protect its host against foreign pathogens by producing antimicrobial peptides and substances [41,42]. Finally, the microbiota is essential in the development of the intestinal mucosa and immune system of the host [43,44].

Factors including extrinsic modulators and host intrinsic factors may cause its wide microbial diversity [45]. External factors include, but are not limited to, host diet, antibiotics, drugs, environmental stressors, exercise/lifestyle, and gastric surgery. Host intrinsic factors include age, gender, genetics, hormones, and bile acids [46,47]. Microbiota composition differs among individuals living in different geographic regions and on different long-term diets [48]. These commensal communities exist in careful balance and, if disrupted, can result in dysbiosis. Dysbiosis is defined as any change to the composition of resident commensal communities relative to the community found in healthy individuals [49]. It can contribute to diseased states such as inflammatory bowel disease, diabetes, and the aetiology of cancers such as colorectal cancer and PDAC [29]. High diversity is a feature of a healthy microbiome, characterised by a large number of beneficial microbes with the ability to resist physiological stress [50,51]. In contrast, in diseased states such as obesity and inflammatory bowel disease, microbiota shows less diversity, with a low number of beneficial microbes and the presence of inflammation-causing pathogens [50,52].

Traditional culture-based methods provide information regarding around 30% of bacterial microbiota [53]. Culture-independent techniques using next-generation sequencing, mainly 16S rRNA gene sequencing, have revolutionised microbial analysis and have enabled the identification and quantification of the many species of microbiota present. It has also enhanced the understanding of how the gut microbiota regulates the innate and adaptive immune homeostasis and the metabolic responses of microbiota to changes in the micro-environment [6,46].

## 4. The Microbiome and Its Relation to Anatomical Site

### 4.1. Oral Microbiome and Its Potential Association with PDAC

Oral health status associated with inflammation of the gingiva, periodontal disease, and tooth loss represents an independent risk factor for PDAC [54,55,56,57]. The exact mechanisms by which oral microbiota reach the pancreas is still unknown. However, the proposed mechanisms involve the translocation via biliary or pancreatic ducts or blood circulation [58,59].

Many studies have demonstrated the role of periodontal pathologies and tooth loss in pancreatic carcinogenesis. A detailed meta-analysis of 49 case-control studies involving 5924 patients found a strong correlation between the presence of *P. gingivalis* and periodontal diseases [60]. Association between the oral pathogens *Porphyromonas gingivalis*, *Fusobacterium*, *Neisseria elongata*, and *Streptococcus mitis,* and PDAC tumorigenesis was evaluated in several studies, with *P. gingivalis* showing a strong positive link with PDAC susceptibility [56,61].

Variations in microbiota in the saliva from patients with PDAC suggested the prospect of using the presence of oral microbes as predictive biomarkers of PDAC [62]. In a prospective case-control study by Fan et al., the authors discovered that the carriage of oral pathogens, *Porphyromonas gingivalis* and *Aggregatibacter actinomycetemcomitans*, were associated with a greater risk of PDAC (odds ratio (OR) for presence versus absence = 1.60 and 95% CI 1.15 to 2.22; OR = 2.20 and 95% CI 1.16 to 4.18, respectively) [61]. On the contrary, the authors also noted that the Phylum Fusobacteria and its genus Leptotrichia were associated with reduced pancreatic cancer risk (OR = 0.94 and 95% CI 0.89 to 0.99; OR = 0.87 and 95% CI 0.79 to 0.95, respectively). In a study by Farrell et al., salivary microbiota from cases with pancreatic cancer were compared with healthy controls. It resulted in the identification of two bacteria, *Neisseria elongate* and *Streptococcus mitis*, and the combination of the two bacterial biomarkers could distinguish cases with PDAC from controls with 96.4% sensitivity and 82.1% specificity [62].

### 4.2. Gut Microbiome and Its Potential Association with PDAC

The gut microbiome is key to the development and modulation of the mucosal innate and adaptive immune system. The human gut microbiota is divided into many different phyla and comprises four main phyla: *Firmicutes, Bacteriodetes, Actinobacteria**,* and *Proteobacteria* [63,64]. Data on the human faecal microbial metagenome showed that an aggregate of 9.9 million microbial genes had been identified [65].

Many studies have established the relationship between risk of gastric cancer and *Helicobacter pylori*, and its role as a risk factor in PDAC has also been evaluated. This includes case-control studies [66,67], prospective cohort studies [68,69] and meta-analyses [70,71]. However, several studies have found no relationship or the opposite [72,73]. Further research is necessary to determine the true impact of *H. pylori* in pancreatic carcinogenesis due to the conflicting results.

Hepatitis B virus (HBV) and hepatitis C virus (HCV) are hepatotropic viruses that can increase the risk of the development of hepatitis and hepatocellular carcinoma (HCC). Hepatitis B virus and HCV infection mainly affect the liver, but they can also be detected in extrahepatic tissues, including the pancreatic juice and the pancreas [74]. Hepatitis B virus and HCV may play a role in the carcinogenesis and the development of extrahepatic malignancies, including PDAC [75,76,77]. Several studies, including several meta-analyses, showed a correlation of HBV infection, including chronic and occult HBV infection, or chronic or inactive HBsAg carries, with increased PDAC risk [78,79,80,81]. According to The Risk Evaluation of Viral Load Elevation and Associated Liver Disease/Cancer-Hepatitis B Virus (REVEAL-HBV) study [82], the association between HBV and PDAC was found in patients with higher viral DNA load (HBV DNA > 300 copies/mL) [82]. The study was designed to investigate the natural history of chronic hepatitis B but made a novel finding in terms of chronic hepatitis B infection with active replication and the increased risk of PDAC.

Studies have also linked the presence of microbes in the bile with an increased risk of PDAC. Serra et al. noted in an observational study involving 53 Italian women that the most frequent disease associated with bactibilia was pancreas head carcinoma (*p* = 0.0015) [83]. The most common microorganisms were *Pseudomonas* spp. (*p* < 0.0001) and *Escherichia coli* (*p* < 0.0001), and these Gram-negative bacteria have been linked to a tumour-associated inflammatory status. Di Carlo et al. subsequently found an unprecedented increase of *E. coli* in bile in 153 patients with pancreatobiliary disease, and this was associated with a decrease in survival [84].

### 4.3. Intratumoral Microbiome and Its Potential Association with PDAC

The mutual interactions between the gut microbiome and the immune system influence the outcome of patients with cancer [44]. However, the interaction between the microbiome inhabiting pancreatic cyst fluid or within the pancreatic tumour is unclear. The presence of bacteria in the PDAC tumour was confirmed in a study by Geller et al. using fluorescence in situ hybridisation (FISH) with fluorescent 16S rRNA-targeted probes and immunohistochemistry using an antibacterial lipopolysaccharide (LPS) antibody [85].

Nejman et al. performed a comprehensive analysis of the tumour microbiome of seven different cancer types, including PDAC [86]. They found that Proteobacteria dominated the microbiome of pancreatic cancer, similar to the normal duodenal microbiome makeup. This may reflect a retrograde bacterial migration from the duodenum into the pancreatic duct. Moreover, in this study, breast and pancreatic tumour samples were found to have the highest proportion of tumours positive for bacterial DNA and were also found to be enriched with Fusobacterium nucleatum compared to other cancer types, including melanoma, lung and ovary [86].

Riquelme et al. performed faecal microbiota transplant (FMT) in mice that later received orthotopic tumour implantation to generate a humanised PDAC microbial mouse model [87]. They demonstrated that about a quarter of the human PDAC microbial composition overlaps with the human donor’s gut microbiota but not the adjacent normal tissue. This suggests microbial cross-talk between the gut and pancreatic tumour. They were also able to detect human donor bacteria in the mice tumour microbiome post-FMT, while it remained absent from mice who did not receive FMT [88]. Faecal microbiota transplant using samples obtained from healthy controls resulted in slower tumour growth and a more modest reversal of tumour micro-environment (TME) immunosuppression than FMT with samples from short-term survivors (STS).

On the other hand, FMT from long-term survivors (LTS) of PDAC resulted in the most potent reversal of TME immunosuppression and tumour growth. This effect was lost when antibiotics were given following the transplant or when cytotoxic T cells were depleted [87]. Figure 1 illustrates the reported microbiome associated with pancreas cancer and its relation to anatomical sites.

## 5. Potential Mechanism of Microbiota in Carcinogenesis

Ten microorganisms have been designated as carcinogens by the International Agency for Cancer Research [89]. This includes *Helicobacter pylori* for its association with stomach cancer. Despite observed links to cancer, a large proportion of these microbes reside within the human population, and many never develop cancers associated with these otherwise commensal microorganisms [6,7,90]. Microbiota can induce carcinogenesis by affecting a number of the cancer hallmarks, such as promoting inflammation, avoiding immune destruction, and microbial metabolites that can de-regulate host genome stability preceding cancer development [6,7,90].

### 5.1. Inflammation

Inflammation is a protective response of the body to harmful stimuli such as pathogens, damaged cells, and toxic compounds [91,92,93]. It acts by removing damaging stimuli and initiating the healing process. However, inflammation can also be a risk factor for cancer development.

A study by Ochi et al. found that Toll-like receptors 4 and 7 (TLR4 and TLR7), which are well-known classes of the pattern-recognition receptors (PRR) family, are up-regulated within the tumour microenvironment of pancreatic cancer [94,95]. Pathogen-associated molecular patterns (PAMPs) are exogenous inflammatory triggers derived from microbial structures which can stimulate an inflammatory response by activating PRRs. Toll-like receptors’ activation can fuel pancreatitis and synergise with KRAS to accelerate pancreatic carcinogenesis in mice. These pro-carcinogenic effects of TLRs can be prevented by inhibiting either nuclear factor kappa-light-chain-enhancer of activated B cells (NF-κB) or mitogen-activated protein kinase (MAPK) pathway [94]. Furthermore, mice deficient in several TLRs are protected from acute pancreatitis. Direct inhibition of TLR4 and TLR7 protects KC mice from pancreatic carcinogenesis [94,95].

Patients with hereditary autoimmune pancreatitis are estimated to carry a lifetime risk of 40% of developing PDAC, and patients with chronic pancreatitis have a 13-fold higher risk of PDAC [96]. The duration of pancreatitis appears to correlate positively with a stepwise increase in the degree of KRAS mutation correlating with the grade of dysplasia of pancreatic intraepithelial neoplasia (PanIN) lesions and eventually leading to PDAC development [97]. This suggests a possible mutagenic role for repetitive bouts of inflammation and the associated genetic change leading to PDAC development.

The microbe can exert not only organ-specific effects, but also through microbe-associated molecular patterns, they can induce pro-inflammatory effects in distant organs. The interaction between bacterial lipopolysaccharide (a PAMP), and host PRR such as TLR4, results in the downstream activation of cell survival pathways, contributing to carcinogenesis outside the GI tract [90,98]. Even though there are a growing number of scientific studies suggesting an underlying infectious component of PDAC aetiology, there is no established infectious source that is clearly carcinogenic for PDAC, such as H. Pylori and its association with gastric cancer [5,61,99,100,101].

### 5.2. Modulation of the Immune System

The relationship between microbiota and carcinogenesis can result in the enhancement of immune response, or the microbiota can exert a pro-tumorigenic response. Some anti-cancer therapy activates the immune system via the gut microbiome and results in an enhanced immune response. For example, Zitvogel and colleagues discovered that cyclophosphamide chemotherapy damages the intestinal mucous layer, allowing gut bacteria to disseminate into lymph nodes and spleen and activate specific immune cells [102]. Additionally, cyclophosphamide lost its anti-cancer effects when given to mice treated with antibiotics or raised free of gut microbes [102,103]. Likewise, Sivan et al. found that Bifidobacterium resulted in an increased response to immunotherapy in mice, suggesting that gut microbiota may activate the immune system [104].

Microbiota can also result in immune response attenuation. Pushalkar et al. discovered that ablation of the microbiome protects against pre-invasive and invasive PDAC, whereas transfer of bacteria from PDAC-bearing hosts, but not controls, reverses tumour protection in mouse models [59]. The combination of an anti-PD1 inhibitor and antibiotics that ablate the microbiota showed a synergistic anti-tumour effect [59].

The heterogeneous findings whereby the microbiome can either enhance or attenuate the immune response to cancer suggest that individual cancer types may induce a distinct alteration of gut and tumour microbiota composition and affect treatment response, including ICI therapy.

### 5.3. Microbial Metabolite and the Regulation of Metabolism

Microbial metabolites play essential roles in numerous biologic and pathologic processes, including translation, gene regulation, stress resistance, cell proliferation, differentiation, apoptosis, and tumour development [105]. Colonic bacteria such as *Faecalibacterium prausnitzii* and *Eubacterium/Roseburia species* have many genes that regulate metabolism and metabolise undigested dietary components that pass through the small intestine [106]. A fibre-rich diet results in saccharolytic fermentation of carbohydrates, leading to the production of short-chain fatty acids (SCFAs), including acetate, butyrate, and propionate [107,108]. Butyrate has been shown to have anti-tumorigenic properties and is associated with decreased CRC incidence [108,109].

On the other hand, proteolytic fermentation with consumption of a meat-rich diet occurs mainly in the distal colon when carbohydrates become depleted and results in the generation of inflammatory and carcinogenic metabolites, such as phenols, ammonia, and other nitrogen-rich metabolites [110]. Diets high in protein and fat content also promote the growth of sulfate-reducing bacteria, such as *Desulfovibrio vulgaris*, and generate excess hydrogen sulfide, which is genotoxic [111,112].

KRAS mutations are characterised as early initiating events [113]. However, oncogenic KRAS alone is not entirely adequate for the development of invasive PDAC. Additional genetic mutations and environmental, nutritional, and metabolic stressors, such as inflammation and obesity, are required for PDAC formation with activation of KRAS downstream effectors. [113]. Evidence in other cancer models has confirmed that dysbiosis induced by a high-fat diet accelerated KRAS-driven intestinal tumorigenesis [114]. There is also evidence that obesity-linked gut microbiota dysbiosis can exert influence on obesity-related cancer, including PDAC [115,116].

## 6. Potential Novel Treatment Approaches for Patients with Pancreatic Ductal Adenocarcinoma

Modulation or restoration of the gut microbiome can be used to treat various gut disorders, including PDAC. Prebiotics, probiotics, antibiotics and FMT are commonly used for gut microbiota modulation [45], and their potential application in the management of patients with PDAC will now be discussed.

### 6.1. Antibiotic Use and its Association with PDAC

Depletion of microbiota with the use of antibiotics may potentially reverse a detrimental dysbiotic environment. For example, eradication of *H. pylori* by the use of a combination of amoxicillin and clarithromycin antibiotics in patients with early gastric cancer is associated with a lower risk of developing metachronous gastric cancer [117].

Pushalkar et al. demonstrated that bacterial ablation using antibiotics in mouse models showed an anti-tumour effect which could be reversed by transferring faeces from PDAC-bearing Pdx1Cre;LSL-KrasG12D;Trp53R172H (KPC) mice. No difference was noted when faeces were transferred from non-PDAC controls. The authors also reported that oral antibiotic administration slowed oncogenic progression; however, select bacterial transfer or bulk faecal transfer from PDAC-bearing mice, but not control mice, resulted in tumorigenesis acceleration [59]. Gut microbiota ablation induces immunogenic reprogramming of the tumour microenvironment and tumour growth suppression by inducing anti-tumourigenic T cell activation, boosting immune surveillance and improving sensitivity to immunotherapy [59,88]. This raised the possibility of using antibiotics in combination with ICIs as an attractive strategy for experimental therapeutics in patients with PDAC and the role of antibiotic chemoprevention in patients with high-risk pancreatic intraepithelial neoplasia.

A retrospective analysis from MD Anderson Cancer Centre (MDACC) reported that macrolide antibiotic use of more than three days during treatment in 148 patients with metastatic PDAC (line of treatment is unclear) that were seen at MDACC led to prolonged PFS and OS [118]. Median PFS for patients taking macrolides was 178 days compared to 124 days in those not taking macrolides (HR = 0.6331, *p*-value = 0.0188). The median OS for patients taking macrolides (*n* = 24) was 541 days compared to 341 days for patients not taking macrolides (n = 144) (HR = 0.6384, *p*-value = 0.0191).

On the contrary, Hasanov et al. [119] reported that tetracycline use in patients with resectable PDAC was associated with significantly shorter OS and a trend to shorter PFS. This retrospective study involved 342 patients with primarily resected PDAC, and antibiotic exposure was seven or more days between diagnosis and surgery. Among the other antibiotics used, tetracycline use was significantly associated with worse survival in patients with resected PDAC. The median OS of patients who had tetracyclines for more than seven days was 687 vs. 1004 days for those not on tetracyclines (HR 1.836; *p* = 0.015).

Microbiota within tumours could confer gemcitabine resistance in patients with PDAC. Certain microbes frequently express cytidine deaminase (CDA) and are capable of converting gemcitabine (2′,2′- difluorodeoxycytidine) into its inactive metabolite 2′,2′-difluorodeoxyuridine. Geller and colleagues [85] assessed PDAC tissue samples obtained during pancreatic surgery for the presence of bacteria. Bacteria were identified in 76% (86 of 113) of PDAC samples compared with only 15% of healthy pancreas samples. Diverse species of intra-tumour bacteria were identified, with members of the Gammaproteobacteria class (which frequently express CDA) being most prevalent. Gemcitabine resistance was conferred by bacteria isolated from 14 of 15 human PDAC tumours [85]. They also found that PDAC cells cultured with a medium that is contaminated with *Mycoplasma hyorhinis* were entirely resistant to gemcitabine [85]. On the other hand, in the MPACT clinical trial involving 430 patients with metastatic PDAC treated with first-line gemcitabine on the comparator arm, antibacterial exposure was associated with an increased risk of gemcitabine-associated toxicity during and after antibiotic exposure (hazard ratio (HR): 1.77; CI: 1.46–2.14) [120].

To date, antibiotic administration is the most aggressive means to manipulate gut microbiota composition, with mixed results so far in PDAC. Other “gentler” options include prebiotic and postbiotic use and will now be discussed.

### 6.2. Probiotic Use and its Association with PDAC

The World Health Organisation has defined probiotics as “live microorganisms which, when administered in adequate amounts, confer a health benefit on the host”. They most frequently belong to the lactic acid bacteria categories *Lactobacillus* spp. and *Bifidobacterium* spp. [121]. Probiotics can be found in processed foods or dietary supplements, including yoghurt, cheese, milk, juices, and smoothies, and fermented foods such as kimchi, kombucha and raw, and unfiltered apple cider vinegar [122].

Several studies have demonstrated the benefits of probiotics in suppressing tumorigenesis, mainly through participating in the innate immune system, decreasing oxidative stress, improving the community of gut microbiota, enhancing intestinal barrier function, and modulating colonisation resistance inhibiting pathogenic bacteria [123,124,125]. However, there are safety concerns in using probiotics in patients with cancer, especially those on immunosuppressive therapies. The risks include bacterial translocation and systemic invasion, and the potential transmission of resistant genes to resident microbiota and the rise of anti-microbial resistance [123].

Dysbiosis of the gut microbiome can increase the incidence of bacterial translocation by modifying intestinal barrier function, which may lead to aggravated inflammation causing chronic pancreatitis and pancreatic cancer. Several animal studies have shown stabilisation of the intestinal barrier by utilising probiotics [126,127,128]. For example, Olah et al. reported that 299 patients with acute pancreatitis treated with *Lactobacillus plantarum* via nasojejunal tube feeding resulted in a lower incidence of pancreatic sepsis, and patients required fewer surgical interventions than control patients [129].

The benefit of probiotics in reducing the risk of development and reducing the risk of relapse has been demonstrated in many cancer types such as colorectal, breast and bladder cancer [130,131,132]. For example, Matsuzaki et al. suggested potent suppression of chemically induced carcinogenesis and anti-metastatic effects on transplantable tumour cells by *Lactobacillus casei* Shirota. The inhibition of tumour growth and increased survival has been observed with the intra-pleural administration of *L. casei* Shirota into tumour-induced sarcoma 180 mice [133]. This process was led by the enhanced production of several cytokines, such as interferon-gamma (IFN-γ), interleukin 1 beta (IL-1β) and tumour necrosis factor-alpha (TNF-α) [134].

There is still insufficient data to draw conclusions about the effects of probiotics for the management of patients with pancreatic cancer, and more work is needed on this topic.

### 6.3. Prebiotic Use and its Association with PDAC

Prebiotics were initially described as “a non-digestible food ingredient that beneficially affects the host by selectively stimulating the growth and activity of one or a limited number of bacteria in the colon, and thus improves host health” [135]. In 2016, the definition was updated as “a substrate that is selectively utilised by host microorganisms conferring a health benefit” [136].

The main prebiotics with health benefits are non-digestible fructooligosaccharides (FOS) and galactans (GOS), preferentially metabolised by *Bifidobacterium* spp. Other examples of prebiotics include polyunsaturated fatty acids (PUFAs) and inulin [136]. Intestinal microorganisms can readily utilise prebiotics, transforming them into metabolic products, such as SCFAs, i.e., propionate, butyrate, and acetate, which are crucial for intestinal health. While colonocytes mainly take up butyrate as primary energy fuel, propionate and acetate are metabolised by the liver and muscle for gluconeogenesis and energy generation, respectively [137].

Prebiotics may act in a probiotic-independent manner and exert a direct effect on the gut. The anti-adhesive properties against pathogens have been studied, and by mimicking the microvillus glycoconjugates, prebiotic oligosaccharides can interact with the bacterial receptor and prevent pathogens from attaching to epithelial cells, thereby inhibiting pathogen colonisation [138,139,140]. To date, there are no published reports on its association with PDAC; however, it may be evaluated in future research.

### 6.4. Synbiotics and Postbiotics

Synbiotics are combinations of prebiotics and probiotics [141]. A previous study has demonstrated that synbiotic supplementation during neoadjuvant chemotherapy for oesophageal cancer improves the gut microbial community and reduces the side effects caused by chemotherapeutic agents [142]. There are no studies using synbiotics in PDAC published as yet.

Postbiotics refer to the soluble by-products and metabolites secreted by gut microbiota that exert host biological activities [143]. A well-known example includes SCFA, which is produced from probiotic fermentation. Postbiotics offer an effective yet safer strategy when compared to the ingestion of viable microorganisms [143]. Postbiotics also offer the potential of protection of the intestinal epithelium and has selective cytotoxicity against a tumour.

Postbiotics remain an unfamiliar area in cancer treatment. There is a substantial diversity of metabolites available, and this presents an enormous challenge for scientists to isolate the molecule for the therapeutic effect and characterise its safety profile in preclinical and clinical settings. This will likely be an area that will continue to develop as a potential cancer therapeutic in the future.

### 6.5. Faecal Microbiota Transplantation

There are several ways to administer FMT, including via endoscopy, rectally as an enema, nasogastric/nasoenteral tube, or oral ingestion of capsules containing faecal material. Faecal microbiota transplant is well established to treat recurrent and refractory *Clostridium difficile* infection, with more than 80–85% efficiency, but not for any other indication [144].

As discussed earlier, Riquelme et al. demonstrated that using FMT in antibiotic-treated mouse models resulted in the gut microbiome colonising the pancreatic tumour and modifying its overall intratumoral bacterial composition [87]. They also found that FMT of the microbiome from the LTS cohort of patients with PDAC induced an anti-tumour response and activation of the immune system in tumour-bearing mice, which was not observed in FMT from patients classified as STS [87]. The tumours from the LTS cohort were characterised by a significant diversity and infiltration with cytotoxic CD8+/killer T cells compared to STS. The findings from this study could propel future research into the use of FMT and microbiome for the treatment of patients with PDAC.

Despite the promising use of FMT, there are potential risks associated with FMT, which include potentially transmitting pathogens to the recipients. Robust microbiological screening is crucial for abolishing this risk, especially of a pathogen with anti-microbial resistance [145].

## 7. The Wider Implications of the Microbiome in PDAC

### 7.1. Cancer Prevention, including PDAC

Identification of the microbiome associated with PDAC provides the potential possibility of its application as a non-invasive diagnostic tool for early PDAC detection using faecal or oral sampling. Such testing has shown promise in colorectal cancer, but its usefulness in PDAC remains unclear [146,147]. Two previously discussed studies by Farrell et al. and Fan et al. showed an association between the oral microbiome and the presence of PDAC [55,61]. Farrell et al. identified that the combination of two microbial biomarkers (*Neisseria elongate* and *Streptococcus mitis)* showed 96.4% sensitivity and 82.1% specificity in differentiating patients with pancreatic cancer from healthy controls [62]. Fan et al. found that the oral pathogens *Porphyromonas gingivalis* (*p* = 0.0048) and *Aggregatibacter actinomycetemcomitans* (*p* = 0.016) were associated with a higher risk of pancreatic cancer [61]. Michaud et al. measured antibodies to oral microbiota in blood samples (before diagnosis) of 405 patients with PDAC and 416 matched healthy controls. They found that patients with high levels of *Porphyromonas gingivalis* antibodies had a twofold higher risk of pancreatic cancer diagnosis (OR 2.14; 95% Cl 1.05, *p* = 0.05) [58]. Detection of relevant oral microbiome appears to be a promising diagnostic biomarker for PDAC; however, its role in PDAC carcinogenesis remains unclear.

Pushalkar et al. noted that oral antibiotic administration slowed oncogenic progression, but select bacterial transfer or bulk faecal transfer from PDAC-bearing mice, but not control mice, resulted in tumorigenesis acceleration and the potential role of antibiotic chemoprevention in patients with high-risk pancreatic intraepithelial neoplasia [59].

### 7.2. The Microbiome and its Potential to Potentiate Efficacy of Anti-Cancer Therapy

The microbiome may also play a role in potentiating the effect of cancer therapy. As discussed earlier, the gemcitabine antitumour effect has been shown to be restored through the ablation of the intrapancreatic microbe using the ciprofloxacin antibiotic in a colon cancer mouse model [85]. Cyclophosphamide efficacy has also been shown to be dependent on the microbiome through an immunological mechanism [102]. Zitvogel and colleagues discovered that cyclophosphamide induces a leaky gut by direct epithelial injury, resulting in increased trafficking of gut microorganisms to the gut-associated lymphoid tissue and activating antitumour Th17 responses. Additionally, cyclophosphamide lost its anti-cancer effects when given to mice (mastocytoma, sarcoma and lung adenocarcinoma mouse models) treated with antibiotics or raised free of gut microbes [102]. Zitvogel and colleagues also explored the influence of gut bacteria in response to anti-CTLA4 and anti-PD1 inhibitors [148,149]. They found that microbe-free mice failed to respond to the ICIs; however, response improved when mice were given *Bacteroides fragilis* [149]. In contrast to gemcitabine, where the ablation of the microbiome restored its effect, removing the gut microbiota by using antibiotics in mice leads to drug resistance to cyclophosphamide [102].

Oxaliplatin, a commonly used platinum agent as part of the FOLFIRINOX regimen, exerts its major therapeutic effect by forming DNA adducts resulting in DNA damage and apoptosis [150]. Iida et al. discovered that a healthy microbiota could enhance the efficacy of the platinum agents by inducing reactive oxygen species’ (ROS) release from myeloid cells, enhancing inflammatory cytokine production and resulting in tumour regression in MC38 colon cancer and B16 melanoma tumour-bearing mice [103]. However, the efficacy of oxaliplatin was impaired when the microbiota was ablated by antibiotics. The absence of resident flora also resulted in a similar reduction in germ-free mice [103]. These results reiterate the potential role of modulation of the microbiome in improving anti-cancer treatment outcomes.

Pushalkar et al. reported that human and mouse PDAC exhibited distinct bacterial compositions, compared with normal pancreas, which suggested potential bacterial translocation from the intestinal tract into the intratumoral environment [59]. They also noted that bacterial ablation with antibiotics reshaped the tumour microenvironment inducing T cell activation, improving immune surveillance and increasing sensitivity to immunotherapy. Pushalkar et al. thus proposed modulating the gut and tumour microbiome as a potential novel strategy to sensitise PDAC tumours to ICI [59]. As the utilisation of immunotherapy in PDAC has been challenging, the microbiome provides an opportunity to sensitise PDAC to immunotherapy.

### 7.3. Alleviating Treatment Side Effects Using the Microbiome

Colitis is one of the well-known adverse effects of ICIs. Metagenomic sequencing revealed an altered profile of certain bacteria in patients who have ICI-induced colitis. The increased presence of bacteria belonging to the Bacteroidetes phylum correlated with resistance to the development of ICI-induced colitis. The identified microbial biomarkers may predict patients’ risk of developing ipilimumab-induced colitis [151]. Bacterial cocktail supplementation to include Bacteroidales and Burkholderiales, and FMT treatment have been shown to ameliorate ICI-induced colitis in antibiotic-treated mice and patients with cancer, respectively [149,152].

Irinotecan hydrochloride (CPT-11) is a topoisomerase-1 inhibitor that is also used as part of the FOLFIRINOX regimen. CPT-11 is converted by liver carboxylesterases into the active metabolite SN-38. SN-38 causes toxicity by damaging the intestinal epithelial cells, and high levels of SN38 can cause side effects such as diarrhoea [153]. Resident microbial β-glucuronidases (GUS) in the intestines can convert the inactive SN-38 G back to the active and toxic SN-38; thus, microbial GUS are believed to be responsible for the side effects of CPT-11 [154,155]. Using a tumour-bearing rat model, Lin et al. demonstrated that CPT-11-based chemotherapy induced microbial dysbiosis in the gut by favouring potentially pathogenic bacteria, such as *Enterobacteriaceae* and *Clostridium* spp. while reducing the presence of beneficial bacteria, such as *Lactobacillus* spp. and *Bifidobacterium* spp. [156]. The side-effects caused by CPT-11 may be reduced by promoting the homeostasis of gut microbiota by enhancing the growth of beneficial gut bacteria and suppression of the pathogenic or opportunistic gut bacteria. 

### 7.4. The Microbiome as a Biomarker in PDAC

The microbiome has the potential to become a prognostic biomarker. Riquelme et al. performed a metagenomic analysis of 68 resected PDAC tumour samples that were compared in two cohorts: STS and LTS [87]. Both cohorts were matched by age, gender, stage, and treatment. The patients classified as LTS had a median survival of 10.1 years and 1.6 years for STS patients. They found that FMT of the microbiome from the LTS cohort of patients with PDAC induced an anti-tumour response and activation of the immune system in tumour-bearing mice, which was not observed in FMT from patients classified as STS [87]. Based on this study, the microbiome may potentially be a future prognostic marker in patients with PDAC.

Zhou et al. analysed faecal microbiota of 32 patients with PDAC, 32 patients with autoimmune pancreatitis and 32 age- and sex-matched healthy controls [157]. They found that alterations in faecal microbiota and butyrate of patients with PDAC suggest an underlying role of gut microbiota in the pathogenesis of PDAC, as well as faecal microbiota and butyrate as potential biomarkers, thus potentially facilitating distinguishing patients with PDAC from patients with autoimmune pancreatitis and healthy controls; however, this requires further validation.

*Fusobacterium nucleatum* may serve as a potential prognostic biomarker for colorectal cancers, but this has not been definitively established in PDAC [158]. In one study, pancreatic tumour samples were found to be enriched by *Fusobacterium nucleatum* [86].

Overall, the microbiome offers numerous potential applications in the wider management of patients with PDAC (Figure 2). Future studies should focus on efforts to elucidate these potentials for patients with PDAC as described in this paper. Table 1 shows some of the ongoing studies involving the microbiome in patients with PDAC. 

## 8. Conclusions

The microbiome lives in enormous abundance and in symbiosis with its human host. A dysbiotic microbiome will undoubtedly affect its human host, either directly or indirectly. It can result in inflammation with ensuing diseased states, modulate the immune response, and regulate metabolism, especially within the gut. The microbiome is also influenced by many factors, including environmental, dietary and genetic variations.

The microbiome represents numerous opportunities for novel therapeutic targets for patients with PDAC; however, this research area is still in its infancy. As discussed earlier in this article in regards to the currently available data on PDAC, a growing body of evidence suggests that the microbiome can influence response to immunotherapy and chemotherapy [85,102,103]. Modulation of the microbiome may alter treatment efficacy, alleviate treatment toxicities, prevent carcinogenesis and potentially has a role as a prognostic biomarker [62,146,149,152,158]. The reciprocal interactions between the gut microbiota and cancer therapies are complicated; it depends on the cancer and therapy type and even the cancer stage. For example, for some cancers such as lung and renal cancer, the presence of gut microbiota is necessary to benefit immunotherapy treatment efficacy [166]; however, in other cancers such as PDAC, depletion of the tumour microbiome can improve treatment response [59].

A very recent paper by Terrisse et al. investigated the influence of the microbiome on the clinical outcome and side effects of early breast cancer treatment [167]. The study showed some similar findings of the potential use of the microbiome as a prognostication biomarker, and the potential of altering breast cancer into a tumour-infiltrating lymphocyte (TIL)-enriched tumour microenvironment, thus making it amenable to ICI therapy. This study [167] also elucidated other potential roles of the microbiome that have not been discussed above concerning PDAC. In this study, they have found other predictive biomarker roles of the microbiome, predictive of neurological treatment side effects and the potential to be utilised as a predictive tool of the efficacy of neoadjuvant and adjuvant chemotherapy [167].

There is still a mammoth task ahead before we can utilise the microbiome in daily clinical use for PDAC management. A substantial amount of exploratory work is required to shape a comprehensive understanding of the microbiome in PDAC. The individual diversity of the intestinal microbiota highlights the difficulty of identifying the most appropriate microbial signatures or targets for pancreatic cancer and poses barriers to this field of research. As a result, and despite the emerging role of the microbiome in PDAC, there remains a paucity in the number of ongoing clinical trials in this field, and a number of studies with interesting findings require further validation. More clinical trials in this area should be encouraged, and future studies should focus on not only the tumour itself but also incorporate the potential impact the microbiome may have on treatment outcomes. Well-designed, controlled, structured, observational and interventional studies of the microbiome in PDAC development and treatment are needed to assess disease associations and the diagnostic and therapeutic potential of the microbiome.

## Figures and Tables

**Figure 1 cancers-13-03779-f001:**
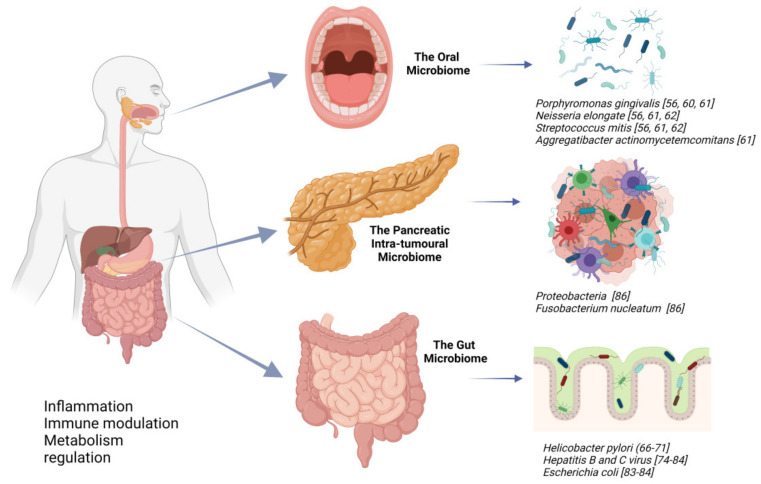
The suggested microbiome related to pancreatic cancer. Created with Biorender.

**Figure 2 cancers-13-03779-f002:**
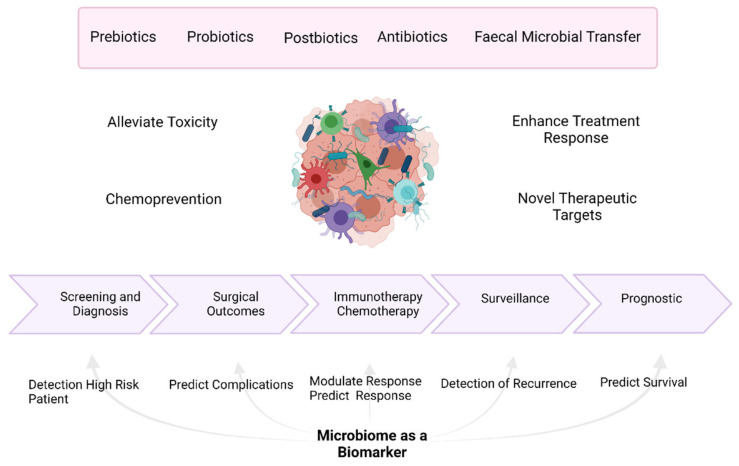
Potential benefits of utilising the microbiome in the management of patients with pancreatic cancer. Created with Biorender.

**Table 1 cancers-13-03779-t001:** Some current studies evaluating the microbiome in patients with pancreatic ductal adenocarcinoma (according to http://clinicaltrials.gov/ in June 2021).

Study	Title	Study Design	Patients (n)	Aim	Study Status	Ref.
NCT03302637	Oral Microbiome and Pancreatic Cancer	A prospective, observational, case-control study	732	To examine the relationship of the oral and pancreatic microbiome and their association with pancreatic cancer risk.	Completed	[159]
NCT04274972	The Microbiome of Pancreatic Cancer: “PANDEMIC” Study	A prospective, observational, cohort study	20	To outline the pancreatic microbiome of patients with resectable PDAC undergoing pancreaticoduodenectomy and characterise the association between microbiome and post-operative complications.	Recruiting	[160]
NCT04189393	Microbiome Analysis in Oesophageal, Pancreatic and Colorectal Cancer Patients Undergoing Gastrointestinal Surgery (MA-PPING)	A prospective, observational, cohort study	60	The study aims to map the oral and gut microbiome of patients diagnosed with pancreatic, oesophageal or colorectal cancer during their surgical patient journey from the moment of diagnosis until full recovery (three months after surgery).	Active, not recruiting	[161]
NCT03840460	A Prospective Translational Tissue Collection Study in Early and Advanced Pancreatic Ductal Adenocarcinoma and Pancreatic Neuroendocrine Tumours to Enable Further Disease Characterisation and the Development of Potential Predictive and Prognostic Biomarkers (PaC-Man)	A prospective observational cohort study	200	This project studies the molecular makeup of pancreatic lesions and their microenvironment at various stages (from pre-cancerous lesions through to more advanced disease) to identify the molecular subtypes, biomarkers of response and toxicity and to investigate the particular intra-pancreatic colonising microorganisms.	Recruiting	[162]
NCT04193904	A Study of Live Biotherapeutic Product MRx0518 With Hypofractionated Radiation Therapy in Resectable Pancreatic Cancer	An interventional open-label phase I study	15	A study to evaluate the safety and preliminary efficacy of MRx0518 with preoperative hypofractionated radiation in 15 patients with resectable pancreatic cancer.	Recruiting	[163]
NCT04600154	MS-20 on Gut Microbiota and Risk/Severity of Cachexia in Pancreatic Cancer Patients	An interventional, randomised study	40	To analyse MS-20 effects on gut microbiota and risk/severity of cachexia in pancreatic cancer patients with a combination of chemotherapy and MS-20.	Active, recruiting	[164]
NCT04203459	The Mechanism of Enhancing the Anti-tumour Effects of CAR-T on PC by Gut Microbiota Regulation	A prospective observational cohort study	80	To study the mechanism of enhancing the antitumor effects of human chimeric antigen receptor T cells on pancreatic cancer by gut microbiota regulation.	Recruiting	[165]

## Data Availability

No new data were created or analyzed in this study. Data sharing is not applicable to this article.

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
