# Peer review of "The Microbiome as a Potential Target for Therapeutic Manipulation in Pancreatic Cancer"

_cancers, 2021, doi:10.3390/cancers13153779_

Round 1

Reviewer 1 Report

This manuscript untitled "The Microbiome As A Potential Target For Therapeutic Manipulation In Pancreatic Cancer" by Rahman et al. is of great interest, well written and brings together new valuable insights in the field of PDAC research. Here are my minor comments about this review:

  • This reference should be added: Montemagno C, Cassim S, De Leiris N, Durivault J, Faraggi M, Pagès G. Pancreatic Ductal Adenocarcinoma: The Dawn of the Era of Nuclear Medicine? Int J Mol Sci. 2021 Jun 15;22(12):6413. doi: 10.3390/ijms22126413. PMID: 34203923; PMCID: PMC8232627.
  • This sentence should be modified to enhance clarity: "Even though there are a growing number of scientific studies suggest an underlying infectious component of PDAC aetiology, there is yet no infectious origins established as carcinogenic for PDAC [5,56,94–96]."
  • This recent article dealing with breast cancer should be added as an example in order to explain how microbiota can be involved/crucial in the clinical outcomes of patients with PDAC.

Terrisse S, Derosa L, Iebba V, Ghiringhelli F, Vaz-Luis I, Kroemer G, Fidelle M, Christodoulidis S, Segata N, Thomas AM, Martin AL, Sirven A, Everhard S, Aprahamian F, Nirmalathasan N, Aarnoutse R, Smidt M, Ziemons J, Caldas C, Loibl S, Denkert C, Durand S, Iglesias C, Pietrantonio F, Routy B, André F, Pasolli E, Delaloge S, Zitvogel L. Intestinal microbiota influences clinical outcome and side effects of early breast cancer treatment. Cell Death Differ. 2021 May 7. doi: 10.1038/s41418-021-00784-1. Epub ahead of print. PMID: 33963313.

  • In my opinion, the Conclusion part is too much general and should be more focused on PDAC: what is known, not know, and finally how microbiota can help patient clinical outcomes based on the observations detailed in the present rewiew. 

Author Response

Reviewer 1

  1. This manuscript untitled "The Microbiome As A Potential Target For Therapeutic Manipulation In Pancreatic Cancer" by Rahman et al. is of great interest, well written and brings together new valuable insights in the field of PDAC research. Here are my minor comments about this review:

I appreciate your kind review and comments.  Thank you.

  1. This reference should be added: Montemagno C, Cassim S, De Leiris N, Durivault J, Faraggi M, Pagès G. Pancreatic Ductal Adenocarcinoma: The Dawn of the Era of Nuclear Medicine? Int J Mol Sci. 2021 Jun 15;22(12):6413. doi: 10.3390/ijms22126413. PMID: 34203923; PMCID: PMC8232627.

I have added the below to “2. Therapeutic Challenges in Treating Patients with Pancreatic Cancer”

Difficulty in early detection of PDAC with the lack of biomarkers and accurate diagnostic radiology tests further contribute to the challenges in treating patients with pancreatic cancer [10–12]. PDAC, even when thought to be diagnosed at early stages, may already have subclinical metastases on currently available imaging modality [13,14]. Recent developments of theranostic nuclear imaging will hopefully improve diagnostic ability [12].

  1. This sentence should be modified to enhance clarity: "Even though there are a growing number of scientific studies suggest an underlying infectious component of PDAC aetiology, there is yet no infectious origins established as carcinogenic for PDAC [5,56,94–96]."

Even though there are a growing number of scientific studies suggesting an underlying infectious component of PDAC aetiology, there is no established infectious source that is clearly carcinogenic for PDAC, such as H. Pylori and its association with gastric cancer [5,56,94–96].

  1. This recent article dealing with breast cancer should be added as an example in order to explain how microbiota can be involved/crucial in the clinical outcomes of patients with PDAC.

Terrisse S, Derosa L, Iebba V, Ghiringhelli F, Vaz-Luis I, Kroemer G, Fidelle M, Christodoulidis S, Segata N, Thomas AM, Martin AL, Sirven A, Everhard S, Aprahamian F, Nirmalathasan N, Aarnoutse R, Smidt M, Ziemons J, Caldas C, Loibl S, Denkert C, Durand S, Iglesias C, Pietrantonio F, Routy B, André F, Pasolli E, Delaloge S, Zitvogel L. Intestinal microbiota influences clinical outcome and side effects of early breast cancer treatment. Cell Death Differ. 2021 May 7. doi: 10.1038/s41418-021-00784-1. Epub ahead of print. PMID: 33963313.

Below was added in the conclusion:

A very recent paper by Terrisse et al, investigated the influence of the microbiome on the clinical outcome and side effects of early breast cancer treatment [165]. The study showed some similar findings of the potential use of the microbiome as a prognostication biomarker and the potential of altering breast cancer into a tumour-infiltrating lymphocyte (TIL)-enriched tumour microenvironment and making it amenable to ICI therapy. This study [165] also elucidated other potential roles of the microbiome that has not been discussed above concerning PDAC. In this study, they have found other predictive biomarker roles of the microbiome, predictive of neurological treatment side effects and the potential to be utilised as a predictive tool of the efficacy of neoadjuvant and adjuvant chemotherapy [165].

  1. In my opinion, the Conclusion part is too much general and should be more focused on PDAC: what is known, not know, and finally how microbiota can help patient clinical outcomes based on the observations detailed in the present rewiew. 

Below were added in the conclusion:

The microbiome represents numerous opportunities for novel therapeutic targets for patients with PDAC; however, this research area is still in its infancy. As discussed earlier in this article in regards to the currently available data on PDAC, a growing body of evidence suggests that the microbiome can influence response to immunotherapy and chemotherapy [85,102,103]. Modulation of the microbiome may alter treatment efficacy, alleviate treatment toxicities, prevent carcinogenesis and potentially has a role as a prognostic biomarker [62,146,149,155,163]. The reciprocal interactions between the gut microbiota and cancer therapies are complicated; it depends on the cancer and therapy type and even the cancer stage. For example, for some cancers such as lung and renal cancer, the presence of gut microbiota is necessary to benefit immunotherapy treatment efficacy [164]; however, in other cancers such as PDAC, depletion of the tumour microbiome can improve treatment response [59].

Reviewer 2 Report

The authors have provided a very good review on the topic.

Minor points are,

  1. A few typos: e.g., page 6 of 27. 2nd line of the 2n paragraph from the bottom. “orthoptic” should be “orthotopic”?
  2. References cited: The formats are mixed and weird in some cases. In a number of references, some key information has been missing:

Ref #1 [(volume? (shorted as “V” from now on)]; 3 (V?); 4 (V and article number?); 7 (V?); 8; 9 (V and article number?); 35 (V?); 39 (V?); 40-44; 47; 48; 52(V?); 67 (article number, not page numbers); 70; 71; #84; 86-88; 102; 104; 106 (article number, not page numbers?); 108; 110; 111; 118-120; 126-128; 130; 133; 135-138; 140; 143 (article number?); 146; 147, etc.

Please provide missing information and use correct formats for all of the references.

Author Response

The authors have provided a very good review of the topic.

Minor points are,

  1. A few typos: e.g., page 6 of 27. 2ndline of the 2n paragraph from the bottom. “orthoptic” should be “orthotopic”?

This typo has been corrected with thanks.

  1. References cited: The formats are mixed and weird in some cases. In a number of references, some key information has been missing:

Ref #1 [(volume? (shorted as “V” from now on)]; 3 (V?); 4 (V and article number?); 7 (V?); 8; 9 (V and article number?); 35 (V?); 39 (V?); 40-44; 47; 48; 52(V?); 67 (article number, not page numbers); 70; 71; #84; 86-88; 102; 104; 106 (article number, not page numbers?); 108; 110; 111; 118-120; 126-128; 130; 133; 135-138; 140; 143 (article number?); 146; 147, etc.

Please provide missing information and use correct formats for all of the references.

I have altered the formatting for references on Mendeley, so all references should now have volume and issue number where applicable as addressed by reviewer #2.